# FINE-TUNING IS ALL YOU NEED TO MITIGATE BACKDOOR ATTACKS

## ABSTRACT

Backdoor attacks represent one of the major threats to machine learning models. Various efforts have been made to mitigate backdoors. However, existing defenses have become increasingly complex and often require high computational resources or may also jeopardize models' utility. In this work, we show that fine-tuning, one of the most common and easy-to-adopt machine learning training operations, can effectively remove backdoors from machine learning models while maintaining high model utility. Extensive experiments over three machine learning paradigms show that fine-tuning and our newly proposed super-fine-tuning achieve strong defense performance. We hope our results can help machine learning model owners better protect their models from backdoor threats. Also, it calls for the design of more advanced attacks in order to comprehensively assess machine learning models' backdoor vulnerabilities.

## 1 INTRODUCTION

In recent years, researchers have shown that machine learning (ML) models are vulnerable to various security attacks. One common attack in this domain is the backdoor attack (Gu et al., 2017; Pang et al., 2020b; Chen et al., 2017; Liu et al., 2018b; Jia et al., 2022; Wu et al., 2022; Shen et al., 2022), whereby an adversary aims to insert a backdoor into a target ML model via data poisoning and/or malicious training. So far, most efforts have gone into the design of effective backdoor attacks against various types of ML models (Gu et al., 2017; Nguyen & Tran, 2020; Chen et al., 2017; Pang et al., 2020a; Li et al., 2021c). To mitigate these attacks, intricate defenses have been proposed. Some of the defenses (Wang et al., 2019; Chen et al., 2019; Liu et al., 2019; Huang et al., 2019; Guo et al., 2019), focus on extracting the trigger from a target ML model via optimization; some aim to detect the inputs with triggers (Tran et al., 2018; Chen et al., 2018; Gao et al., 2019; Udeshi et al., 2022); others rely on training a large set of backdoored shadow models to learn how to differentiate backdoored models from clean ones (Xu et al., 2021).

As defenses become increasingly complex, the defender needs to be equipped with powerful computing infrastructures, which is often a bottleneck. Moreover, to remove the backdoors, some of the defenses need to change the target models' parameters, which jeopardizes the models' performance on the original tasks, i.e., model utility. For instance, one defense named Activation Clustering (AC) (Chen et al., 2018) fails to successfully remove the backdoor (BadNets (Gu et al., 2017)) from the target model trained on CIFAR100 (CIF). Moreover, AC causes the models' accuracy on clean samples to drop from 0.672 to 0.582 (see Section 5.4).

Fine-tuning is a widely adopted technique in the ML training pipeline, especially for transfer learning (Zhuang et al., 2019) and encoder-based learning (Chen et al., 2020; He et al., 2020; Chen & He, 2021; Grill et al., 2020). In this paper, we find that fine-tuning with a proper learning rate is the most effective defense method for mitigating backdoor attacks in terms of both defense performance and utility. Moreover, it is remarkably easy to apply to a variety of machine learning paradigms. Note that we focus on image classifiers as their backdoor vulnerabilities have been extensively studied (Gu et al., 2017; Chen et al., 2017; Pang et al., 2020b).

We consider three types of ML deployment scenarios in this work, namely, an *encoder-based* scenario, a *transfer-based* scenario, and a *standalone* scenario. To measure the performance of backdoor defenses, we consider three metrics, including attack success rate (ASR), model utility (measured by clean accuracy, CA), and computational cost (measured by GPU hours). The former two

are the standard metrics in this field: an effective defense aims to reduce the attack success rate while maintaining the target model's utility. Meanwhile, low computational cost implies the defense can be easily deployed, which is also one of the significant advantages of our approach.

**Methodology.** We empirically show that, in an encoder-based scenario, conventional fine-tuning is sufficient for countering backdoors. In the other two scenarios where conventional fine-tuning is not effective, we further devise *super-fine-tuning*. Our super-fine-tuning method is inspired by super-convergence (Smith & Topin, 2018). We find that a large learning rate significantly helps remove the backdoor, while a small learning rate can maintain the model utility. Therefore, we combine them together and construct a dynamic learning rate method to mitigate backdoor attacks.

**Evaluation.** In the encoder-based scenario, our evaluation shows that the backdoor cannot survive if the user conducts the whole model (conventional) fine-tuning. For instance, when fine-tuning the backdoored encoder trained by BadEncoder (Jia et al., 2022), after one epoch (which takes about 0.004 GPU hours on an NVIDIA DGX-A100 server), the attack success rate on STL10 (STL) (pre-trained on CIFAR10) drops from 0.998 to 0.127. In this scenario, conventional fine-tuning is sufficient. More importantly, it is a **zero-cost** backdoor removal solution, as conventional fine-tuning is a necessary step for users to adapt the pre-trained encoders to downstream tasks (Chen et al., 2020; Kornblith et al., 2019; Li et al., 2020).

In the transfer-based scenario, our experiments show that through conventional fine-tuning, most of the backdoor attacks can be successfully mitigated. On the other hand, our proposed super-fine-tuning can more effectively remove all backdoors with fewer epochs and retain the models' utility. For instance, while conventional fine-tuning can only decrease the ASR from 0.945 to 0.221 on Bad-Nets (Gu et al., 2017) attacks of a CIFAR10 (CIF) model in 100 epochs (about 0.617 GPU hours), super-fine-tuning can make the ASR drop to 0.096 within three epochs (about 0.089 GPU hours) while keeping high utility (0.936). Note that fine-tuning is also necessary for transfer learning to perform downstream tasks; thus, our defense is still costless, similar to the encoder-based scenario.

Normally, the standalone scenario does not need fine-tuning. Here, fine-tuning is an extra step intended to remove the backdoor. However, this does not hurt model utility. In this scenario, conventional fine-tuning does not always work. Instead, by relying on our super-fine-tuning method, we can achieve excellent performance regarding mitigating backdoor attacks. For instance, with 0.089 GPU hours, super-fine-tuning can decrease the ASR of the Blended attack (Chen et al., 2017) on a CIFAR10 model from 0.997 to 0.082 while keeping a high utility (0.937).

To summarize, our experimental results show that in the encoder-based scenario, conventional fine-tuning (on the whole model) is sufficient to remove almost all encoder-based backdoors. For the transfer-based and standalone scenarios, super-fine-tuning can achieve remarkably strong performance.

We compare the performance between super-fine-tuning and other existing state-of-the-art defense methods (Li et al., 2021a; Chen et al., 2018; Liu et al., 2018a; Li et al., 2021b; Wang et al., 2019; Tran et al., 2018). Our results show that super-fine-tuning achieves the best performance in all perspectives (attack success rate, clean accuracy, and computational cost). For instance, the defense method called ABL (Li et al., 2021a) fails in mitigating most of the attacks in the standalone scenario. The ASR of BadNets on CIFAR10 will remain high (0.896) after ABL has been applied. Meanwhile, super-fine-tuning manages to drop the ASR from 0.954 to 0.069.

**Implications.** In general, our results show that backdoor defenses can be performed more easily than previously thought. All one needs is fine-tuning or super-fine-tuning. Currently, the empirical evaluation suggests that backdoor attacks achieve almost perfect accuracy ($\sim$100% accuracy (Pang et al., 2020b; Gu et al., 2017; Chen et al., 2017; Liu et al., 2018b; Jia et al., 2022)), especially for standalone classifiers. By applying our easy-to-deploy fine-tuning defense, our work will certainly help the model users/owners mitigate existing backdoor attacks deployed in the real world. It further calls for the design of more advanced backdoor attacks to better assess the vulnerability of ML models to such attacks.

## 2 BACKDOOR ATTACKS

### 2.1 THE PRINCIPLE OF BACKDOOR ATTACKS

In this work, we focus on targeted backdoor attacks on image classification tasks, which is the most common setting of backdoor-related research. The classification tasks can be formulated as follows: $f(x) = c$, where $x \in \mathbb{X}, c \in \mathbb{C}$. $\mathbb{X}$ is the image domain and $\mathbb{C}$ is the label domain. To inject a backdoor into a target model, an adversary manipulates the model to learn the trigger pattern. Images with this trigger pattern will be classified into the target label. The process can be formulated as the following: $f(t(x)) = c_t$, where $t(\cdot)$ is the pre-defined trigger pattern and $c_t$ is the target label.

Currently, different types of backdoor attacks mainly focus on how to design better trigger patterns (Nguyen & Tran, 2020; Chen et al., 2017; Salem et al., 2022; Liu et al., 2020) or how to improve the backdoor training process (Liu et al., 2018b; Yao et al., 2019; Zhao et al., 2020; Liu et al., 2020; Barni et al., 2019). For better trigger patterns, the adversary aims to bypass the existing defenses to poison the training dataset. With regards to improving the backdoor training process, the adversary aims to inject the backdoor in an easier and faster way. We will introduce the representative attacks in detail in Section 4.1. Our further experiments show that all these backdoor attacks can be easily mitigated by either conventional fine-tuning or our proposed super-fine-tuning method.

### 2.2 ATTACK SCENARIOS

As stated before, we consider three different scenarios in our work, including encoder-based, transfer-based, and standalone scenarios. Based on these scenarios, we recommend users use different fine-tuning strategies (see Section 3 for more details).

**Encoder-Based Scenario.** With the quick development of self-supervised learning, the encoder-based paradigm is becoming popular. The encoder-based paradigm consists of two key steps: pre-training an encoder and constructing downstream classifiers from the encoder for various tasks. Current efforts of the attack mainly focus on injecting backdoors into the encoder and expect downstream classifiers built on the pre-trained encoder to have good backdoor performance as well as high utility. One representative encoder-based backdoor attack is BadEncoder (Jia et al., 2022), where an optimization-based solution is used to train a backdoored image encoder. Concretely, to obtain the backdoored encoder, BadEncoder forces the embeddings of the triggered images to be close to a pre-defined target image's embedding (increasing attack success rate) while keeping clean images' embeddings similar to the corresponding embeddings on the clean model (maintaining model utility).

Normally, backdoor attacks on this encoder-based paradigm assume the users freeze the encoder's parameters and only fine-tune the downstream classifier. In this case, most attacks survive and achieve a high attack success rate as well as high utility. However, in common encoder use cases, the encoder is fine-tuned as well (Tian et al., 2020b), which means that the encoder's parameters are changed too. This may call for extra difficulty in maintaining the attack performance.

**Transfer-Based Scenario.** Another popular scenario is the transfer learning setting, whereby the user gets a pre-trained model on a large-scale dataset (pre-trained model) and then fine-tunes the model to adapt to their own downstream tasks (fine-tuned model). To achieve such adaptation, one common way is to replace the pre-trained model's original classification layer with a new classification layer that fits the downstream task and fine-tune the new model. For backdoor attacks in this scenario, the adversary injects the backdoor in the pre-trained model by associating a trigger with a certain class on a subset of the pre-training dataset. After fine-tuning (with the downstream task dataset), the adversary expects that images with the pre-defined trigger will be misclassified in the fine-tuned model, and the misclassifications all lead to the same (but random) class. We consider this setting as multiple attacks can be easily adapted here like the ones considered in our experiments (Gu et al., 2017; Chen et al., 2017; Zeng et al., 2021; Nguyen & Tran, 2020; 2021). Note that there exists another work on backdoor attacks against transfer learning (Yao et al., 2019). We do not use it as its performance is not strong based on our evaluation as well as the results in (Jia et al., 2022).

**Standalone Scenario.** The most common and difficult scenario is the standalone scenario. In this scenario, the user can directly deploy the model obtained from the Internet without any modification.

Note that the training dataset of the model is usually publicly available to the user. An alternative case is that the user outsources their data to a company that offers ML model training service and then obtains the model from the company (the company being the adversary here). In both cases, the backdoor injected by the adversary makes the model misclassify any inputs with the trigger into the pre-defined class. Our evaluation shows that, even if the user fine-tunes the model with the same dataset that was used to train the backdoored model, the backdoor can still remain, which calls for more effective defenses.

## 3  BACKDOOR DEFENSES

In this section, we first introduce the defender's goals and capabilities. Then, we will discuss how fine-tuning and super-fine-tuning work to mitigate backdoor attacks.

### 3.1  DEFENDER'S GOALS AND CAPABILITIES

**Defender's Goals.** A defender's goal can be summarized from three perspectives.

- **Backdoor Performance.** The main goal of the defender is to reduce the backdoor performance. To achieve this goal, the defender can either detect/mitigate the triggered inputs or purify the model to mitigate the backdoor effect.
- **Utility.** In addition to reducing the backdoor performance, the defender should also keep the utility of the backdoored model. That means that, after the defense, the model should still perform well on clean inputs.
- **Computational Cost.** As ML models become increasingly complex, training and testing models both require one to have powerful computing infrastructures. Ideally, the defender should use minimal computing resources to mitigate backdoors.

**Defender's Capabilities.** The defender is supposed to have a clean dataset to conduct the backdoor defense. For the encoder-based and transfer-based scenarios, this assumption is straightforward. The user (who is also the defender) is the one who fine-tunes the model for their downstream tasks, and they should have the clean dataset already. For the standalone scenario, as mentioned before, the model's training dataset is provided or can be obtained by the user. Moreover, in all the scenarios, we assume the defender has white-box access to the model, which means that they can access and modify the model's parameters. Also, as we have stated before, the defender only has limited computational resources.

### 3.2  FINE-TUNING TO MITIGATE BACKDOOR ATTACKS

In this section, we describe how conventional fine-tuning works and then propose our super-fine-tuning method.

**Conventional Fine-Tuning.** Fine-tuning is a strategy originally proposed in the context of transfer learning. The motivation behind existing fine-tuning is to enable the pre-trained model to fit new data samples using information learned from the pre-training phase. In our case, fine-tuning is supposed to mitigate backdoor attacks as well as leverage the pre-trained model information. Instead of only fine-tuning a few layers like previous works (Jia et al., 2022), we adopt whole model fine-tuning in all our scenarios. During the fine-tuning process, we rely on the same learning rate as the one used in the pre-training process.

In the encoder-based scenario, conventional fine-tuning means that the user conducts the whole model fine-tuning, which is recommended by various existing works (Khosla et al., 2020; Chen et al., 2020; Tian et al., 2020b). Our experimental results show that conventional fine-tuning can effectively mitigate backdoor attacks in the encoder-based scenario, but it does not always work in the transfer-based and standalone scenarios.

**Super-Fine-Tuning.** We further propose a super-fine-tuning strategy, a novel fine-tuning approach focusing on removing backdoor attacks. Super-fine-tuning is inspired by super-convergence(Smith & Topin, 2018), which shows that the regular changes in learning rate can contribute to fast learning. The main innovation of super-fine-tuning is the scheduler of the learning rate. Normally, the gradient

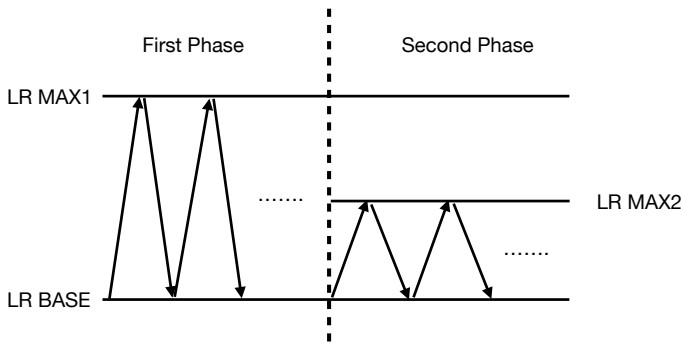

Figure 1: The learning rate scheduler of super-fine-tuning.

descent process can be formulated as $x = x - \epsilon \bigtriangledown_x f(x)$ where $x$ represents the weights of the model, $\epsilon$ represents the learning rate, and $f(\cdot)$ represents the loss function. To make the model forget backdoor triggers while keeping the utility, we make $\epsilon$ change according to the schedule shown in Figure 1. The intuition behind our designed function is that large learning rates tend to make the model forget backdoor triggers while small learning rates maintain the model utility (see Section A.4 for detailed information). Therefore, we combine the two different learning rates with the scheduler.

Concretely, we first pre-define a base learning rate (LR BASE) and two maximum learning rates (LR MAX1 and LR MAX2) for the scheduler of super-fine-tuning. Note that LR MAX1 is required to be larger than LR MAX2. We separate the training process into two phases. In the first phase, we make the learning rate linearly increase from LR BASE to LR MAX1 in several iterations and then drop back to LR BASE. This way, a learning rate that is close to LR MAX1 is supposed to force the model to forget backdoor triggers quickly, while the learning rate that is close to LR BASE will keep the model utility on clean samples. The same process should be repeated until we lower the maximum learning rate after a pre-defined number of epochs (in our experiments, we find that ten epochs work well). In the second phase, we continue oscillating between the base learning rate and LR MAX2 for the remaining epochs, mitigating the overfitting level of the model. Our experimental results show that the above process can effectively mitigate backdoor attacks while retaining the model's utility. Note that we set LR MAX1 as 0.1 and LR BASR as 0.001 according to the super-convergence paper (Smith & Topin, 2018). We set LR MAX2 as 0.01 according to the ResNet paper (He et al., 2016a).

## 4 EXPERIMENTAL SETUP

### 4.1 CURRENT ATTACKS AND DEFENSES

For our evaluation, we consider the following six attacks and six defenses. We introduce them in Supplementary Material in Section A.1. We also introduce our training details in Supplementary Material in Section A.2. For each attack, we set the poison ratio to 0.1.

### 4.2 DATASETS AND EVALUATION METRICS

We leverage five image datasets for our evaluation: CIFAR10 (CIF), CIFAR100 (CIF), STL10 (STL), GTSRB (GTS), and SVHN (SVH) to measure the effectiveness of fine-tuning and super-fine-tuning. To evaluate whether backdoor attacks have been successfully mitigated, we adopt three evaluation metrics following the three goals of the defender described in Section 3.

- **Attack Success Rate.** Attack success rate (ASR) is used to measure whether backdoor samples are successfully classified into the target label or not.
- **Clean Accuracy.** Clean Accuracy (CA) is used to evaluate whether a model can perform well with clean data.

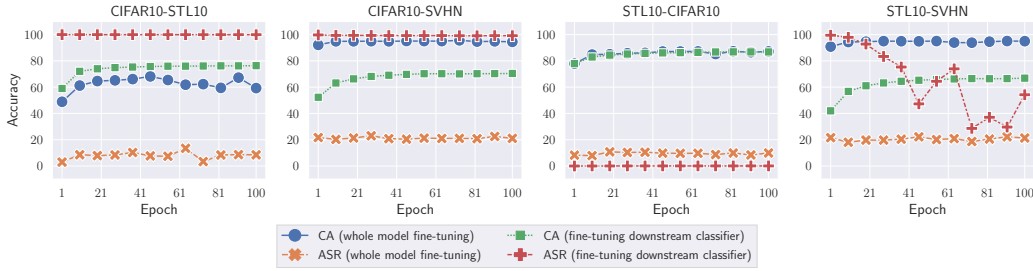

Figure 2: The performance of whole model fine-tuning and downstream classifier fine-tuning on BadEncoder. The X-axis represents training epochs. The Y-axis represents accuracy.

- **Computational Cost.** As we have stated before, when the users take advantage of a third party's pre-trained models, normally, they do not have sufficient computational resources. Therefore, the backdoor defense methods should use as little computational resources as possible. Here, we leverage computational cost (measured by GPU hours) as another new important metric to evaluate defense methods.

## 5 EVALUATION RESULTS

### 5.1 ENCODER-BASED SCENARIO

In the encoder-based scenario, we make use of BadEncoder as backdoor attack since it is the most representative backdoor attack in this setting. The workflow of BadEncoder is to train a backdoored encoder, freeze the encoder, and use the clean data to train a classifier for the downstream task. However, according to previous works (Chen et al., 2020; Khosla et al., 2020; Tian et al., 2020a), fine-tuning the whole model can achieve better performance than only fine-tuning the downstream classifier. Therefore, our fine-tuning method updates the parameters of the whole model.

The experimental results are shown in Figure 2. We train the encoders on CIFAR10 and STL10. Then, we choose CIFAR10, STL10, and SVHN as downstream tasks. From Figure 2, we first observe that BadEncoder is not stable in all datasets. For instance, when the encoder is pre-trained on STL10 and then fine-tuned with CIFAR10, even only fine-tuning the downstream classifier makes the ASR drop to 0.002. Second and more importantly, with whole model conventional fine-tuning, the injected backdoor can always be removed immediately, e.g., within one epoch. For instance, for the encoder pre-trained on CIFAR10 with STL10 as downstream task (shown in Figure 2), when conducting whole model fine-tuning, the ASR drops from 0.998 (fine-tuning downstream classifiers) to 0.127 within one epoch. Note that, in this scenario, whole model conventional fine-tuning is a natural step to achieve better performance on downstream tasks. Therefore, it has **zero-cost** for mitigating backdoor attacks.

### 5.2 TRANSFER-BASED SCENARIO

The transfer-based scenario is also one of the most common machine learning deployment settings. In this scenario, users obtain the model trained on the large dataset and then fine-tune the model on their own dataset to perform the downstream task. We conduct experiments where the backdoored models are pre-trained on CIFAR100 and fine-tuned with CIFAR10 and GTSRB. Here, we adopt five different attack methods described in Section 4.1. As we have stated in Section 2.2, to verify whether a backdoor has been removed, we leverage the original triggers and test whether images with such triggers can be misclassified to a certain class.

The results are shown in Figure 3. As we can see, in this transfer-based scenario, conventional fine-tuning can effectively mitigate backdoor attacks in most cases. For instance, when the defender conducts fine-tuning to the model backdoored by BadNets on CIFAR10, the attack can only achieve 0.378 ASR in one epoch and the ASR will remain around 0.2 after 20 epochs. Our proposed super-fine-tuning method can achieve even better performance than conventional fine-tuning in this scenario. As shown in Figure 3, in most cases, super-fine-tuning can achieve lower ASR with

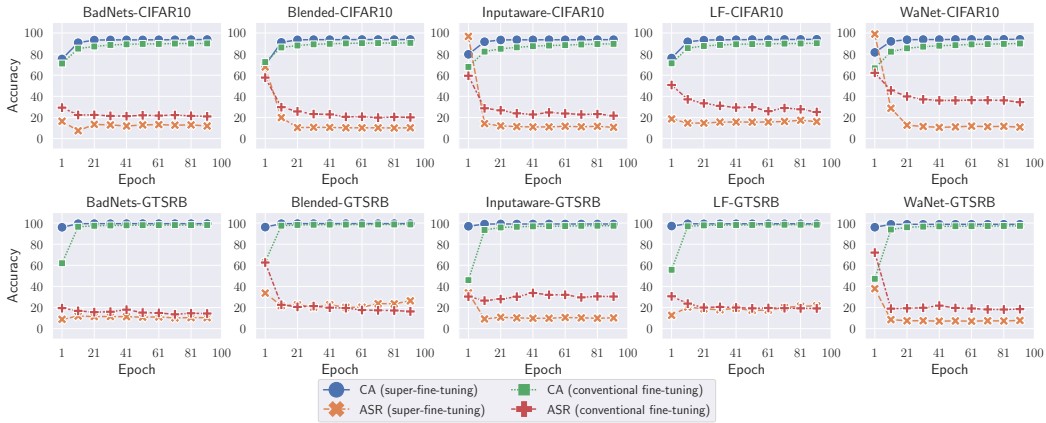

Figure 3: The performance of conventional fine-tuning and super-fine-tuning against different attacks in the transfer-based scenario. The X-axis represents training epochs. The Y-axis represents the accuracy.

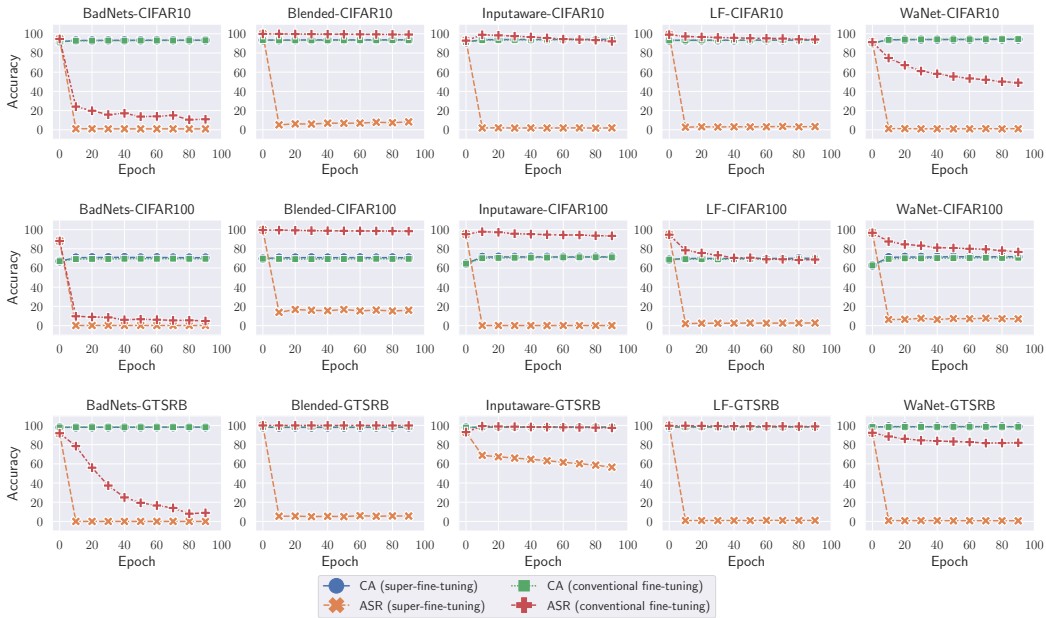

Figure 4: Accuracy of conventional fine-tuning and super-fine-tuning on backdoor samples and clean samples in the standalone scenario. The X-axis represents training epochs. The Y-axis represents the accuracy. Epoch 0 is the original backdoor ASR and CA before fine-tuning or super-fine-tuning.

less epochs. On BadNets-GTSRB, even after the first epoch, ASR will drop to 0.088. Also, it can be seen that super-fine-tuning yields better CA than conventional fine-tuning. For instance, super-fine-tuning on CIFAR10 against Inputaware attacks can achieve 0.798 CA in the first epoch and 0.937 CA after 100 epochs, both higher than conventional fine-tuning (0.678 in the first epoch and 0.898 after 100 epochs). This finding demonstrates that our proposed super-fine-tuning outperforms conventional fine-tuning in this scenario.

## 5.3 STANDALONE SCENARIO

The standalone scenario is the most difficult scenario to mitigate backdoor attacks. Here, a user directly interacts with the model without any modification. Similar to the transfer-based scenario, we adopt the five attacks in Section 4.1. Fine-tuning is no longer a necessary step. Also, due to the

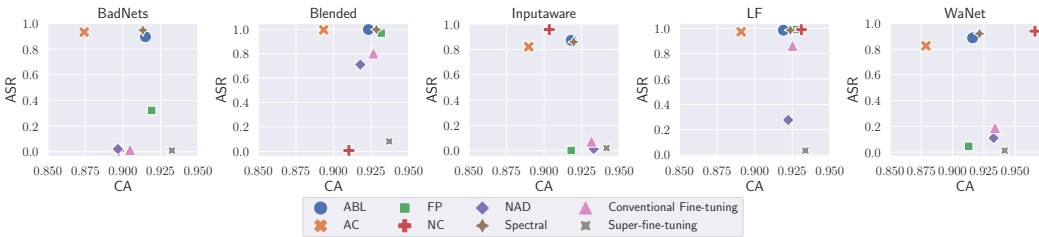

Figure 5: Comparison between existing state-of-the-art backdoor defenses and super-fine-tuning on CIFAR10. The X-axis represents accuracy on clean samples. The Y-axis represents the attack success rate. Points closer to the lower right corner indicate better defense performance.

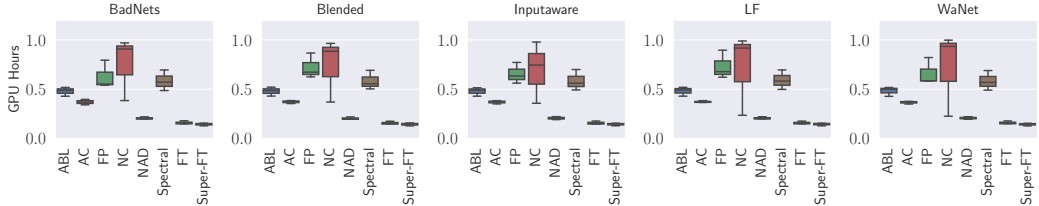

Figure 6: The time cost of different defense methods. The X-axis represents different methods. The Y-axis represents the GPU hours required for this method. Note that in each box, we include the time cost on all datasets.

fact that the model is trained on the desired dataset, it increases the difficulty of mitigating backdoor attacks. Most previous works (Liu et al., 2018a; Gu et al., 2017; Chen et al., 2017) that claim backdoor attacks cannot be easily mitigated by fine-tuning are conducted in this scenario.

As shown in Figure 4, conventional fine-tuning indeed performs poorly in mitigating backdoor attacks in this case. For instance, when conducting conventional fine-tuning on the model backdoored by Blended attacks on CIFAR10, the ASR still remains high (0.978) even after 100 epochs. However, among all five attacks we have studied, super-fine-tuning always decreases the ASR significantly while keeping high clean accuracy. For instance, on CIFAR10, super-fine-tuning can decrease the ASR of Blended backdoor from 0.998 to 0.081, which is in line with the predicted probability of the clean sample. We can also conclude from Figure 4 that super-fine-tuning maintains the model's utility to a large extent. In most cases, the utility does not even drop after the first epoch.

In general, we empirically demonstrate that, with super-fine-tuning, we can effectively mitigate the backdoor attacks while keeping the model utility with a limited number of epochs. Later in Supplementary Material in Section A.4, we will dive into the details of how the learning rate modification affects the ASR and CA.

## 5.4 COMPARISON TO OTHER DEFENSE METHODS

Previously, we have shown that super-fine-tuning can effectively mitigate backdoor attacks with limited computational resources. In this section, we compare super-fine-tuning with other existing state-of-the-art defense methods to show that super-fine-tuning is the most effective and efficient one. Note that here we only focus on the standalone scenario since fine-tuning is a necessary step in the other two scenarios, which means fine-tuning as a defense is zero-cost. Also, fine-tuning or super-fine-tuning can decrease the ASR to a large extent while maintaining the model's utility.

The results on CIFAR10 are shown in Figure 5. We also show the results on CIFAR100 and GTSRB in Figure 7 and Figure 8 in the Supplementary Material. Note that in Figure 5, the X-axis is CA and the Y-axis is ASR. Therefore, in each sub-figure, the closer to the lower right corner, the better the defense performance. Among all defense methods against different attacks, super-fine-tuning, in general, achieves the lowest ASR while maintaining the highest CA. For instance, to mitigate the BadNets attack on CIFAR10, super-fine-tuning can achieve 0.932 CA with only 0.009 ASR, which constitutes the best performance among all defense methods. We can also see that other

defense methods cannot always guarantee performance in defending against all attacks. For instance, although only NC and super-fine-tuning can mitigate Blended attacks on CIFAR10, NC cannot detect Inputaware, LF, and WaNet attacks.

We then consider another important aspect, i.e., each defense's computational cost. The results are shown in Figure 6. We can observe that, among all defenses against different attacks, NC has the largest computational cost, while super-fine-tuning has the lowest computational cost. For instance, to detect and remove BadNets on CIFAR100, NC takes 0.997 GPU hours, while super-fine-tuning only needs 0.147 GPU hours, which is significantly lower.

In general, we conclude that super-fine-tuning outperforms other defenses in terms of the lowest ASR, highest CA, and lowest computational cost. However, from Figure 5, we find that NC costs the largest computational resources. GPU hours needed for NC can even be enough to re-train a new model, which is unacceptable. Compared to NC, super-fine-tuning not only achieves similar or even better performance on removing backdoor, but also costs far less computational resources than NC. It can be seen that among all defending methods, super-fine-tuning costs the least time.

We can also observe that, compared to other methods, the time required for super-fine-tuning is more stable. From Figure 6, we can see that in all attacks and in all datasets used, super-fine-tuning always takes around 0.15 GPU hours. This observation convinces us that fine-tuning will remain costless in almost every setting.

**Takeaway.** In this section, we have shown that fine-tuning is indeed the best way to remove a backdoor in all scenarios. It can be concluded that right now, existing backdoor attacks cannot survive in encoder-based and transfer-based scenarios. In the standalone scenario, with our super-fine-tuning, backdoor attacks can still not be saved, and we have proven that super-fine-tuning is the most effective and costless method.

### 5.5 SUMMARY

In this section, our empirical study shows that backdoor attacks can be easily defended by fine-tuning or super-fine-tuning. Concretely, we find that in the encoder-based and transfer-based scenarios, fine-tuning as the necessary step can naturally remove the existing backdoors. Also, our proposed super-fine-tuning method can better mitigate the backdoor attacks in the transfer-based scenario. In the standalone scenario, super-fine-tuning can effectively prevent backdoor attacks with a limited size of the training dataset and limited computational resources compared to other existing defenses.

## 6 CONCLUSION

In this paper, we have demonstrated that conventional fine-tuning is a very effective backdoor removal method. Moreover, we propose super-fine-tuning which can have even better mitigation performance. We consider three scenarios, namely encoder-based, transfer-based, and standalone. Our experimental results show that in the encoder-based scenario, whole model conventional fine-tuning can effectively remove backdoors within a few epochs. As fine-tuning is a necessary step for users to train downstream classifiers, it can be argued that fine-tuning as a defense method incurs *zero-cost*. In the transfer-based scenario, fine-tuning is still a necessary step. However, we find that conventional fine-tuning cannot always effectively remove all tried backdoor attacks. However, our experimental results show that super-fine-tuning can effectively mitigate backdoor attacks in this scenario. The most difficult scenario is standalone. In this scenario, we assume the backdoored models are trained exactly on users' downstream datasets. We show that even using the same dataset to conduct the fine-tuning, super-fine-tuning can still remove backdoor attacks in a few epochs. We also compare super-fine-tuning with state-of-the-art defense methods and demonstrate that super-fine-tuning outperforms them.

Our results demonstrate that backdoor defenses can be performed in an easier way than previously considered. Fine-tuning or super-fine-tuning is sufficient in most cases. We hope our methods can help ML model owners better shield their models from backdoor attacks. Also, it further calls for the design of more advanced attacks in order to comprehensively assess machine learning models' vulnerabilities to backdoor attacks.

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
