# A  SUPPLEMENTARY MATERIAL

## A.1  BACKDOOR ATTACKS AND DEFENSES

**Attacks.** In this paper, we adopt the following state-of-the-art attacks:

- **BadNets (Gu et al., 2017).** BadNets is the most representative and classic backdoor attack against ML models. The key intuition behind BadNets is to add a visible trigger to some part of the training images and label them with a target class. When the model is trained on this poisoned dataset, the backdoor will be injected, and any inputs with the same trigger will be misclassified into the target class.

- **Blended (Chen et al., 2017).** The Blended backdoor attack is another well-established backdoor attack. Different from BadNets, Blended aims at creating a trigger that is difficult to detect even by human eyes. Also, the position of the trigger does not affect the recognition of the backdoor.

- **LF (Zeng et al., 2021).** The Low Frequency (LF) backdoor aims to design backdoor attacks from a frequency perspective. Previous works' trigger images are significantly different from clean images in the frequency domain. LF aims to make the triggered sample and clean sample consistent in terms of frequency. In this way, backdoor triggers have better concealment in the frequency domain.

- **Inputaware (Nguyen & Tran, 2020).** Inputaware argues that uniform trigger patterns will be easy to detect by simple pattern recognition methods. Therefore, this attack aims to design a generator driven by diversity loss to generate personalized triggers. Through this generator, the triggers for different images are different.

- **WaNet (Nguyen & Tran, 2021).** WaNet also focuses on designing undetectable backdoor attacks. WaNet uses small and smooth deformation technology to generate undetectable trigger samples.

- **BadEncoder (Jia et al., 2022).** Different from previous attacks, BadEncoder conducts backdoor attacks on the encoders (e.g., encoders established by self-supervised learning). It injects backdoors into encoders and then expects the corresponding downstream classifiers to have good backdoor performance as well as high utility.

BadEncoder is designed specifically for the encoder-based scenario, while the other five attacks can be applied to both transfer-based and standalone scenarios. Note that although there are also other similar backdoor attacks on encoders (Liu et al., 2022; Carlini & Terzis, 2022), they share a similar poisoning spirit. Therefore, we only take advantage of BadEncoder for the evaluation.

**Defenses.** Besides fine-tuning and super-fine-tuning, we also evaluate the following six state-of-the-art defense methods.

- **ABL (Li et al., 2021a).** Anti-Backdoor Learning (ABL) aims to train the clean model on the poisoned dataset. The intuition of ABL is that a model tends to remember backdoor samples fast, and backdoor samples are tied to specific classes. ABL designs a two-stage gradient ascent method to isolate backdoor samples and makes the relationship between the backdoor sample and the corresponding label invalid. In this way, ABL can successfully counter backdoor attacks.

- **AC (Chen et al., 2018).** The intuition behind Activation Clustering (AC) is that clean samples and backdoor samples will activate different parameters in neural networks. AC finds the backdoor samples by traversing the parameters of each activation and comparing their distributions.

- **FP (Liu et al., 2018a).** Fine-pruning (FP) is a defense method similar to fine-tuning. However, fine-pruning argues that only conducting fine-tuning cannot effectively mitigate the backdoor attack. Therefore, besides fine-tuning, the method will prune the neural network to eliminate the low influential neurons in order to remove the backdoor in the model. Later we show that proper fine-tuning is sufficient to mitigate backdoor attacks (see Section 5.1).

- **NAD (Li et al., 2021b).** Neural Attention Distillation (NAD) also argues that simply fine-tuning is not enough to mitigate the backdoor attack. They use knowledge distillation with the clean teacher model to guide the fine-tuning process of the backdoored student model. In this way, the backdoor can be removed but the computational cost is high.

- **NC (Wang et al., 2019).** Neural Cleanse (NC) is a classic backdoor detection and removal method. NC optimizes potential triggers in each class and then compares each class' minimum perturbation

Table 1: Super-fine-tuning performance on other attacks.

| | IMC (Pang et al., 2020a) | Latent Backdoor (Yao et al., 2019) | Handcrafted Backdoor (Hong et al., 2022) |
|---|---|---|---|
| Before Super-Fine-Tuning | 0.981 | 0.213 | 0.971 |
| After Super-Fine-Tuning | 0.163 | 0.099 | 0.083 |

to find the out-of-distribution classes. If this class exists, the model is backdoored, and this class is the target class. To mitigate backdoor attacks, NC conducts unlearning by fine-tuning the model using images with triggers and correct samples.

- **Spectral (Tran et al., 2018).** Spectral signatures detection aims to remove triggered samples by detecting the spectrum of the covariance of a feature representation learned by the neural network. Then, the spectral approach will retrain the model with the remaining clean data.

We use BackdoorBench[1] to implement these attacks and defenses. Also, all the experiments are conducted on an NVIDIA DGX-A100 server.

## A.2 TRAINING DETAILS

For the encoder-based scenario, we leverage the official code[2] of BadEncoder to obtain the backdoored encoder. The encoder is ResNet-18 (He et al., 2016a) trained by SimCLR (Chen et al., 2020). We set the batch size to 256 and train the encoder for 1,000 epochs with a learning rate of $3e^{-4}$ and the ADAM (Kingma & Ba, 2015) optimizer. Then, we train the downstream classifier (MLP) by classifier-only fine-tuning and whole model fine-tuning for 200 epochs with a learning rate of $3e^{-4}$. In the transfer-based and standalone-based scenarios, we take advantage of BackdoorBench to train different backdoored attack models. As recommended by BackdoorBench, we insert the backdoor in PreAct-ResNet18 (He et al., 2016b). During the training process, we leverage the SGD optimizer with 0.001 as learning rate, 256 as batch size, and 200 as number of training epochs. For super-fine-tuning, we set LR BASE as $3e^{-4}$, LR MAX1 as 0.1, and LR MAX2 as 0.001. Note that We set LR MAX2 to 0.001 as it's widely adopted by previous work [8,24]. We set LR MAX2 to 0.005 following the instruction of (Shen et al., 2022). For LR MAX1, we already show in Figure 15 that from range 0.1 to 0.5, it doesn't change much. We also set 500 as number of iterations.

## A.3 MORE EXPERIMENTAL RESULTS

Additionally, we also adapt some current state-of-the-art backdoor attacks to evaluate the performance of super-fine-tuning. We show the results in Table 1. It can be concluded that our super-fine-tuning can work very well in almost all tested backdoor attacks. We also provide theoretical proof in the main paper to argue why super-fine-tuning can work so well.

We show more experimental results of comparison to other existing backdoor attacks on CIFAR100 and GTSRB in Figure 7 and Figure 8. We also show more experimental results of backdoor re-injection on different datasets in Figure 9, Figure 10, Figure 11, and Figure 12.

## A.4 ABLATION STUDY

Here, we conduct some ablation studies to show the impact of fine-tuning dataset size and learning rate on the backdoor removal performance. Note that we only focus on the standalone scenario here because: (i) in both encoder-based and transfer-based scenarios, fine-tuning is a necessary step, so we do not modify the fine-tuning dataset size and learning rate; (ii) standalone is the most challenging scenario, as we mentioned before.

**Impact of Fine-Tuning Dataset Size.** We first explore the impact of fine-tuning dataset size. Previously, we used the whole dataset to conduct super-fine-tuning. We have shown that, even with the whole dataset, super-fine-tuning consumes limited computational resources compared to other methods. Then, we further explore how much data is sufficient to conduct a successful super-fine-tuning. We show our experimental results in Figure 13. We can see that even with 20% of the fine-tuning

---

[1] https://github.com/SCLBD/BackdoorBench.
[2] https://github.com/jinyuan-jia/BadEncoder.

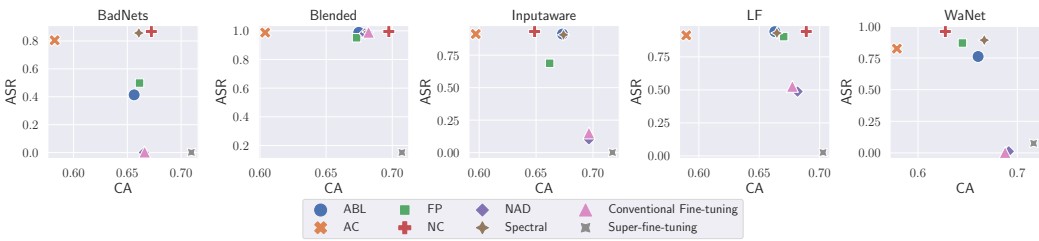

Figure 7: Comparison between existing state-of-the-art backdoor defense methods and super-fine-tuning on CIFAR100. The X-axis represents accuracy on clean samples. The Y-axis represents the attack success rate. Points closer to the lower right corner are better points.

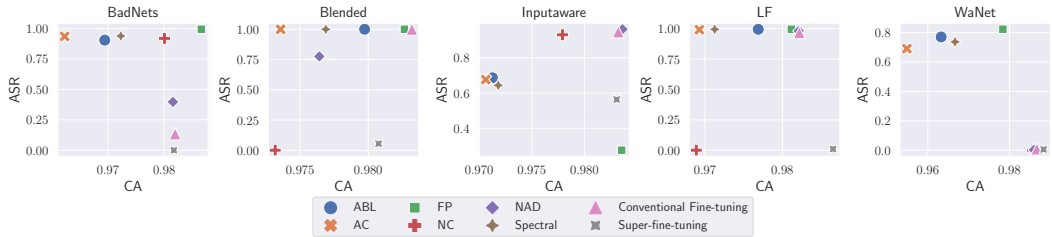

Figure 8: Comparison between existing state-of-the-art backdoor defense methods and super-fine-tuning on GTSRB. The X-axis represents accuracy on clean samples. The Y-axis represents the attack success rate. Points closer to the lower right corner are better points.

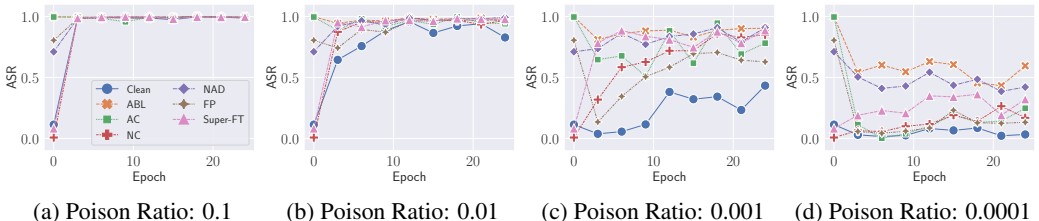

(a) Poison Ratio: 0.1    (b) Poison Ratio: 0.01    (c) Poison Ratio: 0.001    (d) Poison Ratio: 0.0001

Figure 9: Performance of Blended backdoor re-injection attacks on different defense methods. The X-axis represents training epochs in the re-injection phase. The Y-axis represents the accuracy of poison samples.

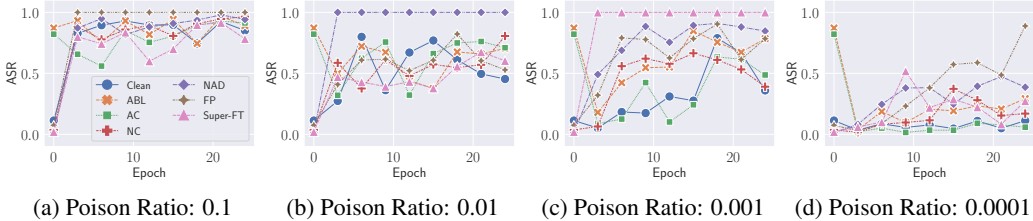

(a) Poison Ratio: 0.1    (b) Poison Ratio: 0.01    (c) Poison Ratio: 0.001    (d) Poison Ratio: 0.0001

Figure 10: Performance of Inputaware backdoor re-injection attacks on different defense methods. The X-axis represents training epochs in the re-injection phase. The Y-axis represents the accuracy of poison samples.

dataset, super-fine-tuning can effectively mitigate the backdoor attacks in most cases. For instance, with 20% of the fine-tuning dataset (CIFAR10), super-fine-tuning reduces the ASR of the Blended backdoor attack to 0.044. Also, from Figure 13, we can see that the size of the fine-tuning dataset has a limited impact on the utility of the model. The clean accuracy remains high with 10% to 100% of the fine-tuning dataset. Therefore, it can be concluded that super-fine-tuning requires significantly

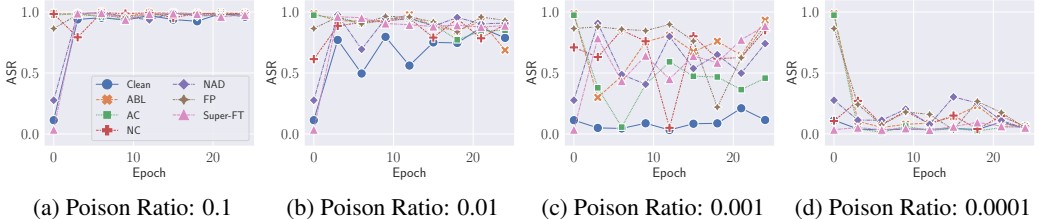

(a) Poison Ratio: 0.1    (b) Poison Ratio: 0.01    (c) Poison Ratio: 0.001    (d) Poison Ratio: 0.0001

Figure 11: Performance of LF backdoor re-injection attacks on different defense methods. The X-axis represents training epochs in the re-injection phase. The Y-axis represents the accuracy of poison samples.

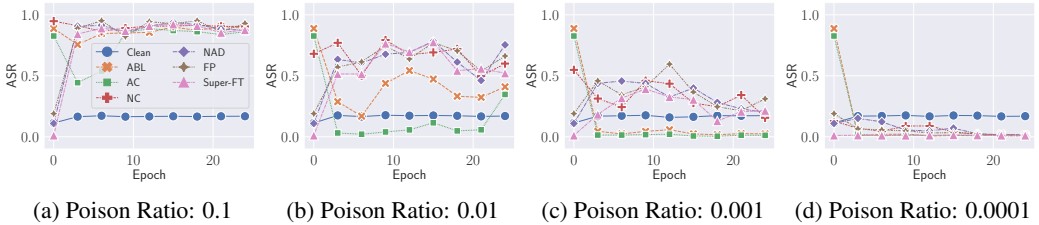

(a) Poison Ratio: 0.1    (b) Poison Ratio: 0.01    (c) Poison Ratio: 0.001    (d) Poison Ratio: 0.0001

Figure 12: Performance of WaNet backdoor re-injection attacks on different defense methods. The X-axis represents training epochs in the re-injection phase. The Y-axis represents the accuracy of poison samples.

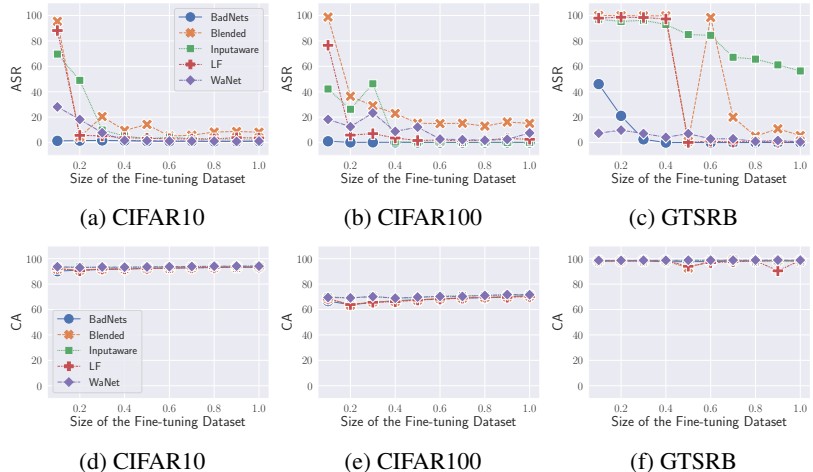

(a) CIFAR10    (b) CIFAR100    (c) GTSRB

(d) CIFAR10    (e) CIFAR100    (f) GTSRB

Figure 13: The impact of fine-tuning dataset size on defense performance. The first row shows fine-tuning dataset size's impacts on attack success rate. The second row shows fine-tuning dataset size's impacts on clean samples accuracy. The X-axis represents the ratio of the fine-tuning dataset, which is used to conduct fine-tuning.

fewer fine-tuning data samples for a stable performance, which further reduces the computational cost.

Note that super-fine-tuning is less effective with 10% of the fine-tuning dataset. For instance, when only using a 10% clean training dataset and 0.1 as LR MAX1, the defender can only achieve 0.954 ASR with the Blended attack on CIFAR10. We show later that the backdoor attacks can still be effectively mitigated by increasing LR MAX1 if the defender only has 10% of the fine-tuning dataset.

**Impact of Learning Rate Change.** During our experiments, we first find that backdoor attacks are very sensitive to different learning rates. We show different learning rates' results of conventional

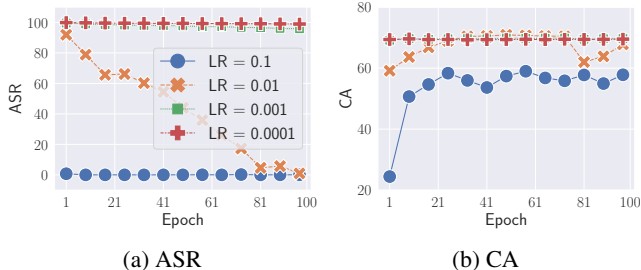

(a) ASR       (b) CA

Figure 14: The impact of different learning rates of conventional fine-tuning on removing backdoor attacks. The X-axis represents training epochs. The Y-axis represents the accuracy of backdoor samples and clean samples.

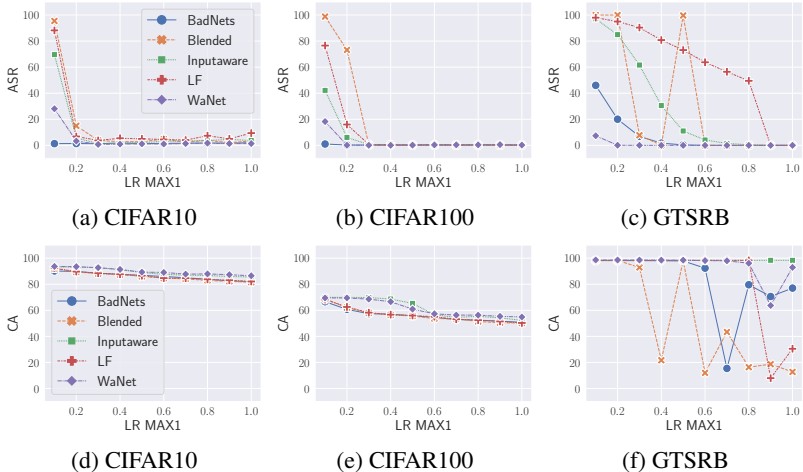

(a) CIFAR10      (b) CIFAR100      (c) GTSRB

(d) CIFAR10      (e) CIFAR100      (f) GTSRB

Figure 15: Impact of LR MAX1 of super-fine-tuning on defense performance. The first row shows LR MAX1's impacts on attack success rate. The second row shows LR MAX1's impacts on clean sample accuracy. The X-axis represents how many data samples are used to conduct fine-tuning. Note that we only use 10% of the fine-tuning dataset to conduct super-fine-tuning.

fine-tuning in Figure 14. It can be seen that if the defender uses the same small learning rate as in the pre-training phase, the ASR remains high even after 100 epochs. However, with an increased learning rate (from 0.0001 to 0.001), backdoor triggers are forgotten gradually in 100 epochs. Moreover, when the learning rate increases to 0.1, the backdoor can be immediately removed within one epoch. We can conclude that learning rates have a significant impact on backdoor removal. In particular, larger learning rates tend to mitigate backdoor attacks faster.

From Figure 14, we can also find that although increasing learning rates can effectively mitigate backdoor attacks, it also causes utility drops. From the utility perspective, small learning rates lead to higher clean accuracy. Therefore, combining large and small learning rates becomes a promising idea to achieve both goals. This is also the general intuition for super-fine-tuning.

In our previous super-fine-tuning experiments, we set LR MAX1 to 0.1 as we find that 0.1 is enough for removing backdoors with sufficient fine-tuning datasets. Here, we also explore the impact of LR MAX1 on super-fine-tuning. To better show the learning rate's impact, we only use 10% of the fine-tuning dataset. In the previous section, when the defender only has 10% of the fine-tuning dataset, super-fine-tuning does not achieve good performance, especially in Blended and LF attacks.

Our experimental results are shown in Figure 15. When increasing the LR MAX1 from 0.1 to 0.3, even when using 10% of the fine-tuning dataset, super-fine-tuning can still successfully remove backdoors. For instance, when the LR MAX1 is 0.1, Blended attacks on CIFAR100 still achieve 0.988 ASR under super-fine-tuning. However, when LR MAX1 increases to 0.3, the ASR drops to

0.004. Although increasing LR MAX1 can more effectively remove the backdoors, it also leads to a small drop in the model's utility. We show these results in Figure 15. For instance, when the learning rate increases from 0.1 to 0.3, the utility of the fine-tuned model from Blended on CIFAR10 drops from 0.919 to 0.881. Therefore, if users have enough clean data, we recommend using 0.1 as the largest learning rate. However, when users only have limited data, increasing the largest learning rate (e.g., from 0.1 to 0.3) also helps mitigate almost all attacks without suffering a large utility drop.