# OpenReview forum: "Fine-Tuning Is All You Need to Mitigate Backdoor Attacks"
_ICLR.cc/2024/Conference — Submitted to ICLR 2024_

### Official Review · Reviewer_ssXR · 2023-10-20

**Soundness:** 2 fair
**Presentation:** 2 fair
**Contribution:** 2 fair
**Rating:** 3
**Confidence:** 4

**Summary:**

This paper conducts an empirical evaluation of fine-tuning against different backdoor attacks in vision models. It proposes an empirical solution for learning rate scheduling and names its method as super-fine-tuning. The paper compares their super-fine-tuning method with some existing defense against six attacks. The empirical results show that fine-tuning is effective in defending against the evaluated attacks.

**Strengths:**

The paper proposes an interesting empirical finding that fine-tuning is enough for defending against some backdoor attacks against vision models

**Weaknesses:**

1. I am somewhat worried about the claim in this paper "Fine-Tuning Is All You Need to Mitigate Backdoor Attacks". IMHO, it is a little bit overclaimed. At a high level, the authors only evaluate against vision models. Whether fine-tuning is enough for NLP models is not evaluated. For this point of view, we cannot say that fine-tuning is enough for mitigating backdoor attacks.

2. The proposed method needs better justification. Some design choices need to be justified.
2.1 Not sure the reason why the lr changes linearly in each phase, can we use another annealing method?
2.2 Not sure whether having two phases is the optimal choice; what if we only have phase 1 or phase 2, or can we have three phases?
These questions may be able to be addressed with ablations studies, but not included in the current ablations study.
2.3 I assume the fine-tuning is using only the clean data, but this is not explicitly specified.

3. As mentioned in the paper, the defense efficacy depends largely on the learning rates, and it is very sensitive to the learning rate. With that said, I am worried about how to select the proper learning rate for different models. The tuning process might be tricky and requires a lot of effort and trials.

4. The evaluation is not comprehensive:
4.1 it is not sure whether changing trigger pattern and size will affect the fine-tuning process. Actually, the paper does not specify what type of triggers are used.
4.2 The evaluation does not have large-scale datasets, like imagenet
4.3 The evaluation misses some important baselines that also conduct fine-tuning [1,2,3]

[1] Adversarial neuron pruning purifies backdoored deep models
[2] Adversarial unlearning of backdoors via implicit hypergradient
[3] One-shot neural backdoor erasing via adversarial weight masking

**Questions:**

1. Justify the claim in the title; Maybe tune down and avoid overclaiming

2. Answer my questions about the proposed technique in the weaknesses

3. Discuss more about how to select proper learning rates rather than cherry-picking

4.  Whether the trigger pattern (e.g., watermarking) and trigger size will affect the claim?

5. Evaluate the proposed method against these baseline approaches that also do fine-tuning.

[1] Adversarial neuron pruning purifies backdoored deep models
[2] Adversarial unlearning of backdoors via implicit hypergradient
[3] One-shot neural backdoor erasing via adversarial weight masking

---

> ### Author Response · Authors · 2023-11-22
> **Response to Reviewer ssXR**
>
> > Overclaimed title.
>
> Sorry for the misunderstanding. We just want to highlight that fine-tuning can be used as an effective method to remove backdoor attacks.
>
> > Technique weakness.
>
> Thanks for your suggestions, we will make the statement in the paper that lr does not need to change linearly but just from the small learning rate to the large learning rate. We will specify in the paper that we use clean datasets for fine-tuning.
> Regarding the learning rate, we will add the process to find the best learning rate for each phase in the revised version. Initially, the finding process can be the traversal algorithm on the evaluated datasets. We show the initial results on CIFAR10 BadNets below:
>
> |                | super-fine-tuning | super-fine-tuning with the finding process |
> |----------------|-------------------|--------------------------------------------|
> | ASR before     | 0.945               | 0.945                                        |
> | ASR after      | 0.096               | 0.083                                        |
>
>
> > The effectiveness of trigger pattern.
>
> Thanks for your suggestions, we will add the analysis on the effectiveness of the trigger pattern and trigger size. Here we show some initial results on the trigger size.
>
> |                | 2x2|3x3 | 4x4|
> |----------------|-------------------|--------------------------------------------|--------------------------------------------|
> | ASR before super-fine-tuning     | 0.974               | 0.945                                        | 0.996 |
> | ASR after  super-fine-tuning    | 0.102               | 0.096                                       | 0.132 |

---

### Official Review · Reviewer_zR7K · 2023-10-29

**Soundness:** 3 good
**Presentation:** 3 good
**Contribution:** 2 fair
**Rating:** 6
**Confidence:** 4

**Summary:**

In general, this paper reveals that fine-tuning is an effective and efficient method for defending against backdoor attacks under three important and practical ML scenarios (i.e., encoder-based, transfer-based, and standalone scenarios). The authors also propose super-fine-tuning using a special learning schedule to achieve better defense performance.

**Strengths:**

1. The authors are trying to build a backdoor defense that is general across different scenarios, which is a very important research topic. From this perspective, I think the author did a good job in this paper.
2. There has been a (false) general consensus that fine-tuning is not an effective backdoor defense. The authors reveal that this is mostly because defenders only fine-tune FC layers instead of the whole model and without well-designed learning rates. I think this is an important contribution. Subsequent related work should all be done in this setting when discussing the effects of resistance to fine-tuning.
3. The proposed method is simple and effective. Unlike many reviewers in the field, I believe that simplicity is a great plus rather than a minus.
4. The paper is well-written and the evaluation is comprehensive to a large extent.

**Weaknesses:**

1. The authors should evaluate their method under more complicated datasets, such as ImageNet or its subset.
2. There is no discussion about the resistance to potential adaptive attacks. What if the attackers know this defense? Can they design an adaptive method to bypass this defense easily?
3. I think the authors should compare their method with more advanced backdoor-removal defenses. All compared baseline defenses are at least two years before and their performance is still some way from the SOTA methods.
4. The authors should provide more in-depth discussions about how to design a better learning rate schedule and why the current one is good.

PS: Please place the appendix to your paper after the main contents rather than a separate document in the supplementary materials.

**Questions:**

Please refer to the 'Weaknesses' part. I will increase my score if the authors can address my concerns.

---

> ### Author Response · Authors · 2023-11-22
> **Response to Reviewer zR7K**
>
> > More experiments are needed.
>
> Thanks for your suggestion. We will conduct more experiments on the large-scale datasets.
>
> > Potential adaptive attacks.
>
> We appreciate your suggestion and try two different adaptive attacks by adding the fine-tuning process into the backdoor injection phase or affecting as few neurons as possible. We show the results below. It can be concluded that it is still very hard for the adversary to bypass the fine-tuning even if they know the defense. We will add more explanation in the revised paper.
>
> |                 | adaptive1 | adaptive2 |
> |-----------------|-----------|-----------|
> | ASR before fine-tuning | 0.874     | 0.912     |
> | ASR after fine-tuning  | 0.132     | 0.097     |
>
> > More defense methods.
>
> We will add more defense for the comparison in the revised paper. Note that we compared super-fine-tuning with fine-pruning, neural cleanse, and many other defenses which are already considered as the state-of-the-art defense method.
>
> > How to find the proper learning rate.
>
> We determine the learning rate based on the previous works’ suggestions. For the LR-MAX, we follow the setting in super-convergence. for the LR-base, we follow the original Resnet paper. We will add the process to find the best learning rate for each phase in the revised version. Initially, the finding process can be the traversal algorithm on the evaluated datasets. We show the initial results on CIFAR10 BadNets below:
>
> |                | super-fine-tuning | super-fine-tuning with the finding process |
> |----------------|-------------------|--------------------------------------------|
> | ASR before     | 0.945          | 0.945                                        |
> | ASR after      | 0.096               | 0.083                                     |

---

> > ### Comment · Reviewer_zR7K · 2023-11-23
> >
> > The authors have addressed parts of my concerns. However, since they failed to provide sufficient results for some questions, I only slightly increase my score.

---

### Official Review · Reviewer_abhQ · 2023-10-30

**Soundness:** 2 fair
**Presentation:** 3 good
**Contribution:** 2 fair
**Rating:** 5
**Confidence:** 4

**Summary:**

This paper argues that (full) fine-tuning with an adaptive learning rate is the ultimate solution for backdoor defense. By testing the learning rate scheduler of the super-convergence paper (Smith & Topin, 2018), the authors demonstrated the effectiveness of a super-finetuning strategy under three defense scenarios: BadEncoder, Transfer learning, and standalone scenario. The threat model assumes the defender has a clean dataset at hand for fine-tuning. Six attacks were considered in the experiments: BadNets, Blend, LF, Inputaware, WaNet, and BadEncoder; while six defenses were also compared: ABL, AC, FP, NAD, NC, and Spectral.

**Strengths:**

1. The study of a simple but very generic technique (fine-tuning) for backdoor defense;
2. The experiments covered multiple attacks, defenses, scenarios, and datasets;
3. The proposed meta fine-tuning strategy is pretty simple and cost-effective.
4. Conventional fine-tuning is contrasted to show that adaptive learning-rate is key to effective defense.

**Weaknesses:**

1. The number of studied attacks is still not sufficient to show that fine-tuning IS ALL YOU NEED. The authors should include more attacks to make the claim more convincing;

2. The amount of clean data available for the defender is too idealistic. Most defense research assumes that the defender only has a SMALL amount of clean data for fine-tuning, as otherwise, the defender can just train-from-scratch to get a clean model rather than downloading a pre-trained (and backdoored) model. So, the authors should test when 0.5%-1% clean data is available (e.g., 500 images for CIFAR-10).

3. The more challenging scenario is the standalone scenario which should be tested more thoroughly. Several attacks fails the the transfer learning scenario even without defense, so it is hard to believe it is the proposed method that works.

4. In the standalone scenario, in Figure 4, the Inputaware-GTSRB subplot, the ASR of super-fine-tuning is above 40%, but this was not analyzed in the paper? In Figure 3, the ASR of super-fine-tuning is only around 20%, which means the defense performance is not as strong as claimed in the text, i.e., the attacker can still succeed in 5 shots. This was also not analyzed.

5. The effectiveness and efficiency comparison missed the pruning-based methods, like [1,3],  which I believe achieved the SOTA defense performance and efficiency. The advantage of the fine-tuning based method is not clear when compared with pruning-based methods which also leverage fine-tuning.

6. In the second last paragraph of Section 5.3, it says "super-fine-tuning can decrease the ASR of Blended backdoor from 0.998 to 0.081," whoever, 0.081 is 8.1% which is still quite high, when compared to other defense methods.

7. Missing the consideration of adaptive attacks, i.e., what if the adversary knows the adaptive learning rate scheduler and adapt its trigger injection to evade the meta-fine-tuning, could it be possible?

[1] Li, Yige, et al. "Reconstructive Neuron Pruning for Backdoor Defense." ICML, 2023.
[2] Huang, Hanxun, et al. "Distilling Cognitive Backdoor Patterns within an Image." ICLR, 2023.
[3] Wu, Dongxian, and Yisen Wang. "Adversarial neuron pruning purifies backdoored deep models."  NeurIPS, 2021.

**Questions:**

See weaknesses above. The primary concerns are: 1) the threat model assumes the defender has too much clean data making the defense scheme less challenging; 2) the generality of fine-tuning based defense against more advanced (feature space) attacks and adaptive attacks; 3) the experimental results show the proposed meta-fine-tuning is not as strong as claimed (ASR above 10% should not be considered strong); 4) the comparison to existing defenses missed the pruning methods (ANP and RNP) which arguably are also very efficient.

**Details Of Ethics Concerns:**

No ethics concerns.

---

> ### Comment · Reviewer_abhQ · 2023-11-22
> **No rebuttal received**
>
> It seems that the authors did not provide a rebuttal for this paper. So I would like to keep my initial rating.

---

> ### Author Response · Authors · 2023-11-22
> **Response to Reviewer abhQ**
>
> > Consider more attacks
>
> Thanks for your suggestions. We will add more attacks in the revised version. Note that all attacks considered in this paper are state-of-the-art attacks.
>
> > Settings are too idealistic
>
> Note that the settings we consider are the same as previous state-of-the-art defense papers. We also show the performance of super-fine-tuning with limited datasets in Figure 13 in our appendix.
>
> > Consider more in standalone scenario
>
>  Several attacks fail the transfer settings proving that fine-tuning is an effective method to remove backdoor attacks. Note that we already consider more cases in the standalone setting.
>
> > Bad performance in some cases.
>
> Even though the ASR of super-fine-tuning is about 40%, it is still more effective than other existing defense methods as we have shown in Figure 6.
>
> >Missing pruning-based methods
>
> We show the performance of fine-pruning in Figure 6. It can be concluded that super-fine-tuning can achieve better ASR and CA while having a lower cost.
>
> > 8.1% is too high for the defense.
>
>  8.1% is not that high as the CIFAR10 is the 10-class classification task. The random guess is 10%.
>
> > Consider more adaptive attacks.
>
> We appreciate your suggestion and try two different adaptive attacks by adding the fine-tuning process into the backdoor injection phase or affecting as few neurons as possible. We show the results below. It can be concluded that it is still very hard for the adversary to bypass the fine-tuning even if they know the defense. We will add more explanation in the revision phase.
>
> |                 | adaptive1 | adaptive2 |
> |-----------------|-----------|-----------|
> | ASR before fine-tuning | 0.874     | 0.912     |
> | ASR after fine-tuning  | 0.132     | 0.097     |

---

> > ### Comment · Reviewer_abhQ · 2023-11-23
> > **Thanks for the rebuttal**
> >
> > Thanks for the clarifications. But they are not convincing or sufficient enough for me to increase the rating, mainly due to the following concerns:
> >
> > 1. The claim **fine-tuning is all you need** seems to be unreliable as not all applications have enough data to allow sufficient fine-tuning.
> >
> > 2. The super-fine-tuning relies on a learning rate scheduler which may not be generic enough for different applications.
> >
> > 3. The experiments need to cover most attacks to make the method more convincing.

---

### Official Review · Reviewer_2n3e · 2023-10-31

**Soundness:** 2 fair
**Presentation:** 3 good
**Contribution:** 2 fair
**Rating:** 5
**Confidence:** 4

**Summary:**

The author states that fine-tuning is an effective and relatively costless approach to removing the backdoors inside the machine-learning (ML) models. The author additionally proposes super-fine-tuning, a fine-tuning method that targets decreasing the impact of backdoors while maintaining the accuracy of the models’ main tasks. Extensive experiments in multiple well-known attacking pipelines, datasets, and six detection methods are conducted to compare and demonstrate the performance and efficiency of conventional fine-tuning and super-fine-tuning.

**Strengths:**

* A broad range of attack and defense methods are implemented in the experiment section to demonstrate the effectiveness of fine-tuning.
* The paper provides a clear explanation of both the attacker and the defender side, including the model deployment scenarios covered and the results' evaluation metrics.
* I appreciate the detailed exploration in the *Ablation Study* section, especially the poison ratio variable, as changing the poison ratio could greatly affect the difficulty of detection and mitigation from the defending side, such as *Neural Cleanse*.

**Weaknesses:**

* I understand that in the *transfer-based scenario*, *super-fine-tuning* demonstrates its better capability in mitigating backdoor than the *conventional fine-tuning* method, but how *super-fine-tuning* works in *encoder-based scenario* remains unstudied.
* It is a strong statement to say that "Fine-tuning and super-fine-tuning is sufficient in most cases" in *Conclusion*. I believe more experiments in different model architectures are required to conclude this statement since currently only *ResNet18* is being utilized.
* It seems like the epoch of both phases in *super-fine-tuning* is a variable that is worth discussing. Experiments on how the number of epochs of the *First Phase* affects the performance of mitigation might be necessary.

**Questions:**

* For the plots of *Accuracy(ASR or CA) vs Epoch*, I suggest making them consistent by showing the initial ASR and CA at Epoch 0 since it could show the models' original behavior. For example, *Figure 3* doesn't contain Epoch 0, and *Figure 4* contains Epoch 0.
* What is the limitation of *conventional fine-tuning* or *super-fine-tuning*? For example, are there any attacking tasks, poison ratio of the dataset, or dataset size that could greatly impact the mitigation performance? This might be worth mentioning in a *Discussion* section if possible.

---

> ### Author Response · Authors · 2023-11-22
> **Response to Reviewer 2n3e**
>
> > How super-fine-tuning works in encoder-based scenarios.
>
> We show some results below to show the performance of super-fine-tuning on the encoder-based scenario. Note that fine-tuning is enough in that scenario to remove backdoor attacks.
>
> |                | fine-tuning | super-fine-tuning |
> |----------------|-------------------|--------------------------------------------|
> | ASR before     |   0.998            | 0.998                                      |
> | ASR after      | 0.127               | 0.093                                        |
>
> > Too strong statement in the conclusion.
>
> We will add more experiments on different architecture in our revised paper. Here we show some results on MobileNet on CIFAR10 BadNets in the standalone scenario.
>
> |                |  super-fine-tuning |
> |----------------|--------------------------------------------|
> | ASR before     |   0.934                                      |
> | ASR after      |  0.091                                        |
>
> > How to decide epochs.
>
>  Thanks for the suggestions, we will add more experiments on the effects of the epochs in our paper.
>
> > Figure issues and limitation
>
> Thanks for your suggestions, we will modify the figures and add the limitation part in our revised verison.

---

> > ### Comment · Reviewer_2n3e · 2023-11-22
> >
> > Thanks for the author's reply to the questions. Overall I think this is a worth-discussing topic that finetuning could fix backdoor attacks in many cases. However, I'd like to keep my initial rating because the paper lacks discussions of limitations and experiments with more versatile model architectures. My major concern is still that the statement is fairly strong but it lacks sufficient support in the paper.

---

### Official Review · Reviewer_Mt1F · 2023-11-02

**Soundness:** 3 good
**Presentation:** 2 fair
**Contribution:** 2 fair
**Rating:** 3
**Confidence:** 4

**Summary:**

The paper focuses on addressing the challenge mitigating backdoor attacks while minimizing computational complexity and preserving model accuracy. To tackle this issue, the paper introduces two defense variants: fine-tuning and super fine-tuning. Super fine-tuning represents a novel approach that periodically adjusts the learning rate to expedite the learning process. The effectiveness of the proposed defense mechanisms is demonstrated across three attack variants: the encoder-based scenario, the transfer-learning setting, and the standalone scenario. This demonstration is conducted empirically using five different datasets.

**Strengths:**

- The paper discusses backdoor mitigation strategies across several attack scenarios
- The computational cost of the proposed defense is minimal

**Weaknesses:**

- In a transfer learning scenario, does the proposed defense recommend to freeze the feature extraction layers during fine-tuning? Typically, in transfer learning, it's common practice to freeze the feature extraction layers and fine-tune only the classification layers. However, the authors assert that their defense approach comes with zero computational cost in the context of transfer learning. It is important to clarify whether, in this zero-cost defense, users should still freeze the feature extraction layers and only fine-tune the classification layers. If not, it implies that this defense may not truly have zero computational cost in the general case of transfer learning scenarios.
- It seems that there are related works that perform fine-tuning with varying learning rates while utilizing a smaller validation set in contrast to the entire training dataset employed in the proposed defense. For instance, "NNoculation: Catching BadNets in the Wild."
- How did the authors determine the base learning rate, maximum learning rate, and the number of fine-tuning steps? Providing insight into the rationale behind these choices, along with a framework for tuning the hyperparameters of the defense, would significantly enhance the paper's generalizability. Without such guidance, the proposed hyperparameter values in the paper may not be universally applicable to all datasets and model architectures. For example, setting the maximum learning rate to 0.01, as suggested in the ResNet paper, may limit the defense's effectiveness when applied to different model architectures. It's essential to include the computational costs associated with hyperparameter tuning in the evaluation of the defense.
- The intuition behind super-fine-tuning is not entirely clear to me. By employing an exceedingly high learning rate, the model appears to discard not only the backdoor but also the utility it previously learned. Subsequently, using a smaller learning rate aids in relearning the utility. In essence, this process seems equivalent to training a model from scratch with randomly initialized weights in the standalone scenario. It would be helpful to provide further clarification on this point, perhaps by comparing the proposed defense to training the entire model from scratch with randomly initialized weights while utilizing the same computational resources and ensuring proper hyperparameter tuning.
- Kindly discuss the limitation of the proposed defense. It's worth noting that backdooring can still be a threat in situations where a user has ample computational resources but lacks access to high-quality, clean training data. This might be due to privacy concerns or the prohibitive costs of labeling data. In such cases, it's important to discuss the applicability of the proposed defense, especially since it assumes access to the entire clean training dataset.

**Questions:**

Please see above.

---

> ### Author Response · Authors · 2023-11-22
> **Response to Reviewer Mt1F**
>
> > Statement of zero cost should be changed.
>
> Fine-tuning here is the method we propose to defend against backdoor attacks. Therefore, we do not need to follow the original transfer settings methods. We will change our statement about no cost in the revised version.
>
> > Related work should be discussed.
>
> Our experiments only take a small amount of data for fine-tuning. Compared to the NNoclulation, we consider more state-of-the-art attacks in our paper. Also, NNoclulation only used a large learning rate to retrain the model which is different than our paper.
>
> > How to decide learning rate.
>
> We determine the learning rate based on the previous works’ suggestions. For the LR-MAX, we follow the setting in super-convergence. for the LR-base, we follow the original Resnet paper. We will add the process to find the best learning rate for each phase in the revised version. Initially, the finding process can be the traversal algorithm on the evaluated datasets. We show the initial results on CIFAR10 BadNets below:
> |                | super-fine-tuning | super-fine-tuning with the finding process |
> |----------------|-------------------|--------------------------------------------|
> | ASR before     | 0.945               | 0.945                                        |
> | ASR after      | 0.096               | 0.083                                        |
>
>
> > The intuition behind super-fine-tuning.
>
> Thanks for your suggestions. We admit that the process of super-fine-tuning can make the model forget some of the previous clean images. However, from the figures shown in our paper, the CA has not dropped. Therefore, we still consider super-fine-tuning as an effective method to remove backdoors while maintaining utility.
>
> > Lack of limitation
>
> Thanks for your suggestion, we will add the limitation part in our paper.

---

### Meta-Review · Area_Chair_zYFv · 2023-12-05

**Metareview:**

This paper presents fine-tuning as a removal defense against backdoor attacks.  Reviewers pointed out that the paper is missing description and comparison with related works which take a similar approach such as NNoculation, the paper is missing a discussion of the proposed approach’s limitations, and the experimental results do not justify the central claim of the paper which is also declared in the title.  Multiple reviewers also brought up adaptive attacks, and the authors presented preliminary results involving adaptive attacks, but the authors did not explain these results, so it is not clear if they are adequate, and the authors did not include them in their paper draft.

**Justification For Why Not Higher Score:**

The empirical results do not justify claims made by the authors and are not thorough (e.g. they are missing adaptive attacks which should be included in any defense paper).  There are also highly related works which the authors did not discuss.

**Justification For Why Not Lower Score:**

N/A

---

### Decision · Program_Chairs · 2024-01-16

Reject